**Data Availability Statement:** The dataset used in this research uses harmonised data from three cohort studies. The original and the harmonised

# Duration of obesity exposure between ages 10 and 40 years and its relationship with cardiometabolic disease risk factors: A cohort study

**Tom Norris**[1]*, **Tim J. Cole**[2], **David Bann**[3], **Mark Hamer**[4], **Rebecca Hardy**[5], **Leah Li**[2], **Ken K. Ong**[6], **George B. Ploubidis**[3], **Russell Viner**[2], **William Johnson**[1]

**1** School of Sport Exercise and Health Sciences, Loughborough University, Loughborough, United Kingdom, **2** UCL Great Ormond Street Institute of Child Health, London, United Kingdom, **3** Centre for Longitudinal Studies, Department of Social Science, University College London, London, United Kingdom, **4** Division of Surgery & Interventional Science/Institute of Sport, Exercise and Health, UCL, London, United Kingdom, **5** UCL Institute of Education, London, United Kingdom, **6** MRC Epidemiology Unit, University of Cambridge, Cambridge, United Kingdom

* t.norris@lboro.ac.uk

## Abstract

### Background

Individuals with obesity do not represent a homogeneous group in terms of cardiometabolic risk. Using 3 nationally representative British birth cohorts, we investigated whether the duration of obesity was related to heterogeneity in cardiometabolic risk.

### Methods and findings

We used harmonised body mass index (BMI) and cardiometabolic disease risk factor data from 20,746 participants (49.1% male and 97.2% white British) enrolled in 3 British birth cohort studies: the 1946 National Survey of Health and Development (NSHD), the 1958 National Child Development Study (NCDS), and the 1970 British Cohort Study (BCS70). Within each cohort, individual life course BMI trajectories were created between 10 and 40 years of age, and from these, age of obesity onset, duration spent obese (range 0 to 30 years), and cumulative obesity severity were derived. Obesity duration was examined in relation to a number of cardiometabolic disease risk factors collected in mid-adulthood: systolic (SBP) and diastolic blood pressure (DBP), high-density-lipoprotein cholesterol (HDL-C), and glycated haemoglobin (HbA1c).

A greater obesity duration was associated with worse values for all cardiometabolic disease risk factors. The strongest association with obesity duration was for HbA1c: HbA1c levels in those with obesity for <5 years were relatively higher by 5% (95% CI: 4, 6), compared with never obese, increasing to 20% (95% CI: 17, 23) higher in those with obesity for 20 to 30 years. When adjustment was made for obesity severity, the association with obesity duration was largely attenuated for SBP, DBP, and HDL-C. For HbA1c, however, the association with obesity duration persisted, independent of obesity severity. Due to pooling of 3

1946 NSHD data are made available to researchers who submit data requests to mrclha.swiftinfo@ucl.ac.uk; see also the full policy documents at http://www.nshd.mrc.ac.uk/data.aspx. The original data for the 1958 NCDS and 1970 BCS are available from the UK Data Service (https://ukdataservice.ac.uk/); applications for access to any data (original or harmonized) held by the UK Data Archive that forms part of the NCDS Biomedical Resource will require special license and should be submitted to clsfeedback@ioe.ac.uk.

**Funding:** WJ is supported by a UK Medical Research Council (MRC) New Investigator Research Grant (MR/P023347/1) (https://mrc.ukri.org/). WJ acknowledges support from the National Institute for Health Research (NIHR) Leicester Biomedical Research Centre, which is a partnership between University Hospitals of Leicester NHS Trust, Loughborough University, and the University of Leicester. DB is supported by the Economic and Social Research Council (grant number ES/M001660/1) (https://esrc.ukri.org/) and The Academy of Medical Sciences / Wellcome Trust ("Springboard Health of the Public in 2040" award: HOP001/1025) (https://wellcome.ac.uk/). RH is Director of the CLOSER consortium which is funded by the Economic and Social Research Council (award reference: ES/K000357/1) (https://esrc.ukri.org/). TJC was supported by MRC research grant MR/R010692/1 (https://mrc.ukri.org/). GBP is supported by the Economic and Social Research Council ES/M001660/1 (https://esrc.ukri.org/). The funders had no role in study design, data collection and analysis, decision to publish, or preparation of the manuscript.

**Competing interests:** The authors have declared that no competing interests exist.

**Abbreviations:** AUC, area under the curve; BCS70, 1970 British Cohort Study; BMI, body mass index; BIC, Bayesian information criterion; BP, blood pressure; DBP, diastolic blood pressure; FMI, fraction missing information; HbA1c, glycated haemoglobin; HDL-C, high-density-lipoprotein cholesterol; IOTF, International Obesity Task Force; IQR, interquartile range; MICE, multiple imputation by chained equations; NCDS, National Child Development Study; NSHD, National Survey of Health and Development; RR, relative risk; SBP, systolic blood pressure; SEP, socioeconomic position; STROBE, Strengthening the Reporting of Observational Studies in Epidemiology.

cohorts and thus the availability of only a limited number harmonised variables across cohorts, our models included adjustment for only a small number of potential confounding variables, meaning there is a possibility of residual confounding.

## Conclusions

Given that the obesity epidemic is characterised by a much earlier onset of obesity and consequently a greater lifetime exposure, our findings suggest that health policy recommendations aimed at preventing early obesity onset, and therefore reducing lifetime exposure, may help reduce the risk of diabetes, independently of obesity severity. However, to test the robustness of our observed associations, triangulation of evidence from different epidemiological approaches (e.g., mendelian randomization and negative control studies) should be obtained.

## Author summary

### Why was this study done?

- People with obesity (body mass index (BMI) > 30 kg/m$^2$) do not all share the same risk for development of cardiometabolic disease risk factors.

- The duration a person has spent with obesity over their life course could be 1 factor contributing to the variation observed in cardiometabolic risk.

- However, previous studies have been unable to adequately separate the effects of obesity duration (how long a person has been obese) and obesity severity (the magnitude of a person's BMI).

### What did the researchers do and find?

- We derived BMI trajectories between 10 and 40 years of age in 20,746 participants and calculated each person's total time spent with obesity (duration) as well as their severity of obesity.

- We related obesity duration to cardiometabolic disease risk factors (SBP and DBP, HDL-C, and HbA1c) in mid-adulthood.

- A greater obesity duration was associated with worse values for all cardiometabolic disease risk factors. The strongest association with obesity duration was for HbA1c: HbA1c levels in those with obesity for <5 years were relatively higher by 5% (95% CI: 4, 6), compared with never obese, increasing to 20% (95% CI: 17, 23) higher in those with obesity for 20 to 30 years.

- This positive association between obesity duration and cardiometabolic disease risk factors was largely attenuated when adjusting for obesity severity, except for HbA1c.

**What do these findings mean?**

- The obesity epidemic is characterised by trends towards earlier onset and consequently greater lifetime exposure.

- Our findings are important as they suggest that health policy recommendations aimed at preventing early onset obesity, and therefore reducing lifetime obesity exposure, may help reduce the risk for diabetes.

- However, due to the small number of potential confounding variables we were able to include in our analysis, the contribution of residual confounding to our findings should be acknowledged. Furthermore, the robustness of the observed associations should be tested using different markers of glucose metabolism and triangulated using different epidemiological approaches underpinned by different assumptions and sources of bias (e.g. mendelian randomization and negative control studies).

## Introduction

Obesity is a global public health concern. Worldwide prevalence of child and adolescent obesity (defined according to a body mass index [BMI] of >2 standard deviations above age-specific World Health Organization cut-offs) has increased from 0.9% and 0.7% in boys and girls, respectively, in 1975 to 7.8% and 5.6%, respectively, in 2016. These increases in child obesity accompany significant increases in global adult obesity, with prevalence increasing from 3% and 6.6% of males and females, respectively, in 1975 to 11.6% and 15.7%, respectively, in 2016 (defined according to a BMI of >30 kg/m$^2$) [1]. While this epidemic is associated with many adverse health outcomes, particularly cardiovascular disease-related morbidity and mortality [2], individuals with obesity do not represent a homogeneous group in terms of cardiometabolic risk. Indeed, there exists a group of individuals who, while exceeding the standard BMI cut-off for obesity ($\geq$30 kg/m$^2$), are regarded as metabolically healthy because they have an absence of other major cardiovascular risk factors. The life course traits contributing to this heterogeneity in cardiometabolic risk have received little attention, but it is likely that a large proportion of the heterogeneity is related, in particular, to the length of time a person spends obese [3,4]. It has been demonstrated that younger individuals are now accumulating greater exposure to overweight or obesity throughout their lives [5], so a comprehensive understanding of the influence of the duration of obesity on the development of cardiometabolic risk factors is critical.

Abraham and colleagues [6] published one of the first studies investigating this heterogeneity in cardiometabolic risk for a given weight, observing that rates of some cardiovascular diseases were highest among individuals who were most overweight in adulthood but below the average weight in childhood. As this study, and others which have replicated that analysis [7–10], are based on weight status at just 2 time points however, obesity duration can be estimated only crudely. More frequent longitudinal measurements of weight are required for a fuller picture. Furthermore, a detailed measurement schedule is also required in order to differentiate between obesity duration, the age of obesity onset, and the severity of obesity, as these, although correlated, may confer different health risks [11,12]. For example, due to the changes in insulin sensitivity that occur during pubertal development [13], an obesity onset in

adolescence may be more deleterious for insulin resistance and diabetes than an onset during another period of the life course.

A handful of studies with such data have observed positive associations between obesity duration and several cardiometabolic disease risk factors including metabolic syndrome [11,14], hypertriglyceridemia [14], dyslipidaemia [14,15], and blood pressure [16]. Most evidence relate to the association with type 2 diabetes, however, with numerous studies observing a positive relationship with obesity duration [15,17–23]. The largest of these studies ($n$ = 61,821) [21] observed that for each 2-year increment in obesity duration, the risk of type 2 diabetes increased by 14%, although, as observed in other studies [19,22], estimates were attenuated upon adjustment for current weight (representing obesity severity). However, these studies have important limitations, including retrospective designs [14,15], categorising the outcome variable (thus ignoring the observed distribution) [14,15,17,20,21], an a priori assumption of a linear relationship between obesity duration and outcomes [17,18], and assuming that once a person becomes obese they remain obese, thus removing the possibility for weight cycling [14,15,21–23]. Another important limitation is the adjustment of the obesity duration–outcome relationship for current BMI (i.e., at outcome assessment) in order to separate the effects of obesity duration and severity [15,18,20–22]. BMI at outcome assessment does not capture the true extent of obesity severity as it ignores (potentially greater) severity occurring at earlier time points. For example, consider 2 adults, adult A and adult B, who both have a BMI of 35 kg/m$^2$ at follow-up and who have both been obese for 20 years. Adult A has had a constant BMI of 35 kg/m$^2$, while adult B has had a BMI as high as 45 kg/m$^2$ during this period. It is unlikely that the cardiometabolic health risks associated with these 2 profiles are homogeneous.

Recently, the concept of "obese-years" has been proposed, which combines the degree and duration of obesity into a single variable [24,25]. In the study by Araujo and colleagues [25], an area under the curve (AUC) of BMI (BMI$_{AUC}$) was used to summarize duration and severity of BMI. It is possible however to obtain a mean obesity severity over any period by dividing this AUC by obesity duration, thus separating the effects of severity and duration. To our knowledge, no study has done this, and thus, robust evidence of the association between obesity duration and cardiometabolic risk factors, which is truly independent of obesity severity, is lacking.

Using data from 3 British birth cohort studies, the aim of the present study was to model serial measurements of BMI obtained across the life course in order to derive, for each individual, the following obesity traits: duration of obesity exposure between ages 10 and 40 years, age of obesity onset, and obesity severity. These parameters were then used to relate obesity duration, with and without adjustment for obesity severity, to systolic (SBP) and diastolic blood pressure (DBP), high-density lipoprotein cholesterol (HDL-C), and glycated haemoglobin (HbA1c) in mid-adulthood.

## Methods

### Samples

The 3 British birth cohort studies used in these analyses have been previously described in detail elsewhere [26–28] and were designed to be nationally representative when initiated. The MRC National Survey of Health and Development (NSHD) was initiated in 1946 and recruited 5,362 participants. The National Child Development Study (NCDS) was initiated in 1958 and recruited 17,416 participants. The 1970 British Cohort Study (BCS70) was initiated in 1970 and recruited 16,571 participants.

## Ethics statement

All of the studies have received ethical approval and obtained informed parental and/or participant consent, both of which cover the secondary analyses reported here. Data collection in the NSHD received multicentre research ethics committee approval (MREC98/1/121), the NCDS obtained ethical approval from the South East MREC (ref:01/1/44) and the BCS70 obtained ethical approval from the National Research Ethics Service (NRES) Committee South East Coast–Brighton and Sussex (Ref. 15/LO/1446). Further details are available from the study websites and/or cohort profiles [26–29].

For this analysis, we identified a target sample of 20,746 (NSHD: $n$ = 2,968; NCDS: $n$ = 9,302; BCS: $n$ = 8,476) participants who attended the biomedical sweep where cardiometabolic disease risk factor data were collected (see below) and contributed BMI data for the derivation of our exposure variable: obesity duration (S1 Fig).

## Serial BMI data

As described elsewhere [30], serial BMI (kg/m$^2$) was derived and harmonised in each study from measured or self-reported weight and height collected at the target ages 11, 15, 20 (self-report), 26 (self-report), 36, and 43 years in the 1946 NSHD; 11, 16, 23 (self-report), 33, and 42 (self-report) years in the 1958 NCDS; and 10, 16 (one-third self-report), 26 (self-report), 30 (self-report), 34 (self-report), and 42 (self-report) years in the 1970 BCS.

There were 21,009 observations of BMI from 4,702 participants in the 1946 cohort, with 74% of the sample having 4 or more observations. There were 57,545 observations of BMI from 16.274 participants in the 1958 cohort, with 80% of the sample having 3 or more observations. Finally, there were 56,275 observations of BMI from 15,437 participants in the 1970 cohort, with 72% of the sample having 3 or more observations.

## Cardiometabolic disease risk factors in adulthood

In each cohort, a biomedical sweep, with venous blood sampling was conducted in adulthood, at 53 years in the 1946 cohort ($n$ = 3 053), 44 years in the 1958 cohort ($n$ = 9 377), and 46 years in the 1970 cohort ($n$ = 8 581). Measurements of SBP and DBP were obtained as well as blood cardiometabolic biomarkers (HbA1c and HDL-C). More information about the measurement protocols can be found in S1 Text.

## Statistical analysis

TN and WJ determined which analyses to perform and include in the present paper in January 2019 after discussing options with all coauthors. The analysis plan was revised in May (modelling obesity duration as a categorical variable rather than a continuous variable) and October 2019 (removing LDL-cholesterol as an outcome due to high amount of missing data) when further exposure and outcome data were obtained and explored. Further analyses were added in June 2020 in response to reviewer comments (adjusting for further putative confounding variables in the regression models, adding *sexXduration* interaction models in the supplementary analyses).

**Obesity duration parameters.**   In order to identify obesity and derive obesity parameters throughout the life course, we modelled individual child-adulthood trajectories of BMI from 10 to 40 years of age. These life course BMI trajectories were modelled within each cohort separately, due to the previously described between-cohort heterogeneity in the age-related progression of obesity from childhood to adulthood [5]. Models included all participants who contributed at least 1 measurement of BMI during the studied age range (NSHD: 11 to 43

years; NCDS: 11 to 42 years; BCS: 10 to 42 years). The BMI trajectories were modelled using restricted cubic splines with mixed effects, with measurement occasion at level 1 and individuals at level 2. The restricted cubic splines split the trajectories into piecewise functions of age separated by "knots." Between the adjacent knots, separate cubic polynomials were fitted, with the spline terms constrained to be linear in the 2 tails. The number of knots (using the default knot positions as proposed by Harrell [31]) was chosen based on the Bayesian information criterion (BIC), with a lower BIC indicating a better fitting model. Once the best fitting model was identified, sex was added as a fixed effect and as interaction terms with the age terms identified in the previous step. Finally, an adjustment for level 1 variation was included to allow for differing error associated with measured versus self-reported BMI. From these models, fitted annual BMI values between 10 and 40 years were obtained for each individual.

Using these fitted BMI values, z-scores were created relative to the International Obesity Task Force (IOTF) reference [32]. Obesity was defined as a z-score of $>2.288$ in males and $>2.192$ in females, which corresponds to a BMI value of 30 kg/m$^2$ at 18 years. Using the sex-specific obesity cut-off, several obesity parameters were derived for each individual. Firstly, the presence of obesity at any timepoint was identified, representing any BMI z-score, which exceeded the obesity threshold. Secondly, the "number of times obese" was calculated as the number of times an individual's BMI z-score crossed upwards through the obesity threshold. Thirdly, "age first obese" was derived, representing the age, in years, when BMI z-score first crossed upwards through the obesity threshold. "Total duration of obesity" was calculated as the length of time, in years, that a person's BMI z-score exceeded the obesity threshold; these values were categorised as 0: never obese; 1: obesity 0.01 to $<5$ years; 2: obesity 5 to $<10$ years; 3: obesity 10 to $<15$ years; 4: obesity 15 to $<20$ years; 5: obesity 20+ years. Finally, we used the composite trapezoid rule to derive a cumulative obesity severity variable, represented in Fig 1 by the AUC and above the obesity threshold. Severity here is expressed in BMI-years above the obesity threshold, reflecting the fact that it incorporates both duration of obesity and the extent

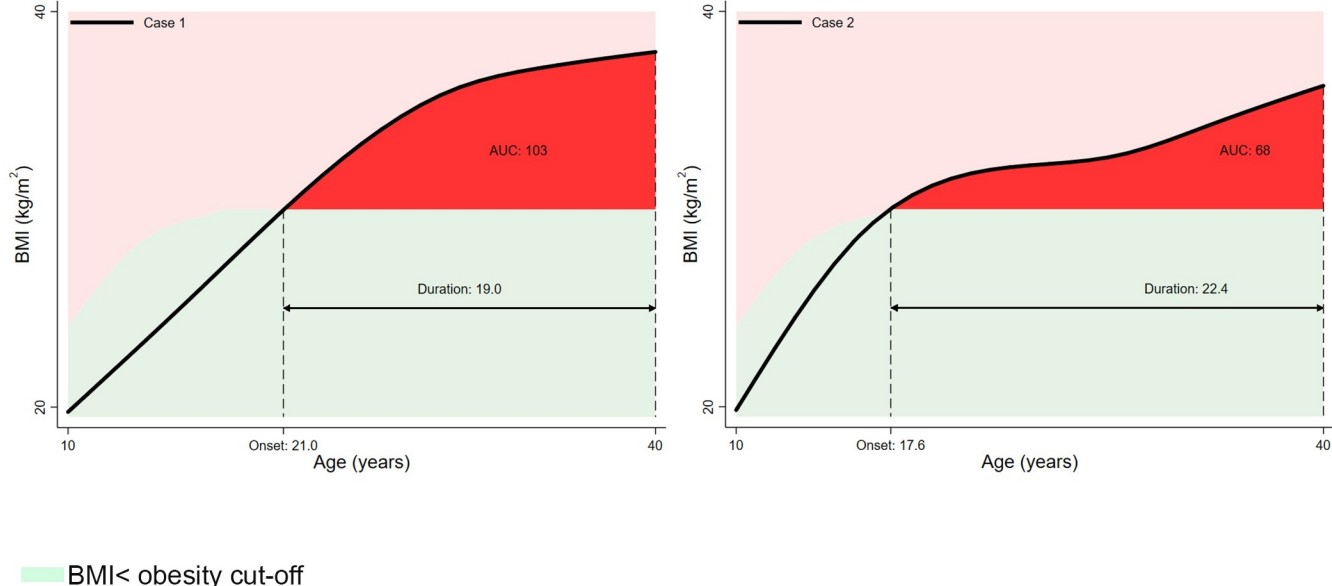

**Fig 1. Example obesity traits (onset, duration, and severity (AUC and above obesity cut-off)) derived from the BMI-z-score trajectories of 2 random participants.** AUC, area under the curve; BMI, body mass index.

to which BMI exceeded the age-specific obesity threshold. If this is then divided by obesity duration, it can be interpreted as the "average obesity severity," i.e., the mean excess BMI above the obesity cut-off.

**Linking obesity parameters to cardiometabolic disease risk factors.** Prespecified constants were added to the cardiometabolic disease risk factors to adjust for being on medication, which has been found to reduce bias [33,34]. The constants were +10 mmHg and +5 mmHg for SBP and DBP, respectively, −5% for HDL-C and +1% (absolute) for HbA1c, obtained from meta-analyses of the effect of blood pressure lowering [35], lipid-regulating [36–38], and diabetes [39] medications on the respective cardiometabolic risk factors.

Multiple linear regression was used to relate obesity parameters to the continuous cardiometabolic risk factors. As uncertainty in estimated obesity parameters are not taken into account in the confidence intervals for their associations with these continuous cardiometabolic risk factors, standard errors may be underestimated. To correct for this, robust standard errors in these subsequent models were estimated. Data were pooled across cohorts and sexes, thus enabling adjustment of the association between obesity duration and cardiometabolic risk factors for cohort and sex. As HDL-C and HbA1c required transformation to achieve normal distributions, for consistency, we transformed all continuous cardiometabolic risk factors to the 100 $\log_e$ scale, so that the regression coefficients are in units of percentage difference in cardiometabolic risk factor per unit difference in covariate [40]. In a first set of models, the binary variable ever (versus never) obese (between 10 and 40 years) was tested for association with each cardiometabolic risk factor. In a subsequent set of models, we related the categorical obesity duration variable to each cardiometabolic risk factor, with never obese the referent group. The above steps were unadjusted for covariates. A subsequent model included adjustments for sex, cohort, birth weight (kg), ethnicity (white versus non-white), social class in childhood (father's social class reported when the child was 10 to 11 years and according to the Registrar General's Social Classes schema; see S2 Text for more details), and age at follow-up. A final model also included an adjustment for average obesity severity. In order to address missingness in covariate data, we used multiple imputation by chained equations (MICE) [41] combining estimates using Rubin's rules [42]. The number of imputations required to achieve convergence of parameter estimates was determined as 100 × fraction missing information (FMI) [43].

In addition, in order to aid presentation, we repeated the above steps for a number of derived dichotomous cardiometabolic disease risk factor variables, using generalised linear models (Poisson distribution with robust error variances) to estimate relative risks (RRs) for each outcome. The derived cardiometabolic disease risk factor variables were hypertension (SBP > 140mmHg and/or DBP > 90mmHg or reported use of BP-lowering medication), low HDL-cholesterol (<1.03mmol/L in males and <1.29mmol/L in females [44] or reported use of lipid-regulating medication) and elevated HbA1c (>5.7% [45] or reported use of diabetes medication).

Beta coefficients from these regression models, i.e., percentage change for continuous variables and RRs for binary variables, were plotted. Each figure is split into 2, with the left-hand side (model 1) showing the estimates from the regression of ever obese (versus never) and the right-hand side (model 2) showing estimates of the categorical obesity duration variable (versus never).

**Sensitivity analyses.** First, we repeated the analyses excluding the NSHD cohort as the biomedical sweep occurred much later in this cohort compared with NCDS and BCS70 cohorts, which may have resulted in an underestimation of the association between obesity duration and cardiometabolic disease risk factors. In a related sensitivity analyses, we also replaced the NSHD blood pressure variables to those collected at the 43-year sweep in order to

align with the timing of blood pressure measurements in the NCDS and BCS70. No other out-come data were available at that age in NSHD however. To identify the extent to which rela-tionships were sex-specific, we also repeated the analyses including a "sex X obesity duration" interaction. We also performed an analysis, which was restricted to those who remained obese, assuming that relationships would strengthen when limited to those with persistent obesity and not cycles of obesity.

Analyses were performed in Stata version 15 (Stata Corp, College Station, Texas) and R ver-sion 3.5.3 (R Core Team 2019, Vienna, Austria).

This study is reported as per the Strengthening the Reporting of Observational Studies in Epidemiology (STROBE) guideline (S1 Checklist).

### Code availability

The statistical code for the analyses in this paper has been placed in GitHub, the open-access online repository (repository URL: https://github.com/tomnorris1988/Obesity-duration-and-cardiometabolic-outcomes).

## Results

Descriptive statistics of the cohorts are shown in Table 1. A total of 49.1% of the sample were male, and 97.2% were white British. As shown in Table 1, the prevalence of "ever obese" between 10 and 40 years was approximately 3 times higher in the most recent cohort BCS70 (19.7%, $n$ = 1673), compared with the oldest cohort NSHD (6.6%, $n$ = 197). The average age of first obesity onset was less in more recent cohorts, with a median of 30.2 years (interquartile range [IQR]: 25.2, 34.1) in the BCS70 compared with 33.4 years (IQR: 27.6, 37.0) in the NSHD. Accordingly, the duration of obesity was greater in the most recent cohort BCS70: median 9.7 years (IQR: 5.9, 14.7), compared with NSHD: 6.2 years (IQR: 2.7, 11.8). The nega-tive correlation between age of obesity onset and duration of obesity was almost perfectly colinear in the more recent cohort (BCS: −0.99; NCDS: −0.95; NSHD: −0.81), indicating almost universal persistence of obesity following its onset in BCS70. Fig 1 provides examples of the BMI-z-score trajectories and the derived obesity parameters.

The average BMI at the biomedical sweep was in the overweight category ($>25$ kg/m$^2$) in all 3 cohorts but was highest in the BCS70 cohort (27.6 kg/m$^2$; IQR: 24.6, 31.5). For all 5 cardi-ometabolic risk factors, the mean was highest in the NSHD cohort, reflecting the older age at follow-up. This was most notable for SBP, with a mean of 136.0 mmHg (SD: 20.1) in the NSHD compared with 126.6 mmHg (16.5) and 124.6 mmHg (15.2) in the NCDS and BCS70, respectively. This translated to a much higher prevalence of hypertension in the NSHD cohort (68.1%) compared with the NCDS (27.8%) and BCS70 cohorts (23.9%). The presence of ele-vated HbA1c was also considerably higher in the NSHD cohort compared with the NCDS and BCS70 (35.8% versus 15.0% and 16.5%, respectively).

### Relationship of obesity parameters with cardiometabolic disease risk factors

Results from the unadjusted analysis are included in S1 and S2 Tables. Here, we report esti-mates from the adjusted analyses, presented in Figs 2–4 and with corresponding estimates in S3–S6 Tables.

**HbA1c.** Being ever obese at any age between 10 and 40 years (versus never obese) was associated with a 9.0% (95% CI: 8.2, 9.9) higher HbA1c (Fig 2, left panel), which reduced to 4.5% higher (95% CI: 3.5, 5.6) when adjusted for obesity severity. HbA1c increased linearly with obesity duration, from 5% excess for $<5$ years duration up to 19.9% (95% CI: 16.5, 23.3) for 20 to 30 years duration ($p$(trend) $< 0.001$). Upon adjustment for obesity severity, the trend

**Table 1. Descriptive statistics for life course obesity parameters and cardiometabolic disease risk factors at the biomedical sweep of those in target study sample (n = 20,746).**

| | | NSHD 1946 (n = 2,968) | | 1958 NCDS (n = 9,302) | | 1970 BCS (n = 8,476) | |
|---|---|---|---|---|---|---|---|
| **Sex** | | | | | | | |
| *Males* | n (%) | 1,459 (49.2) | | 4,630 (49.8) | | 4,106 (48.4) | |
| *Females* | n (%) | 1,509 (50.8) | | 4,672 (50.2) | | 4,370 (51.6) | |
| **Ethnicity** | | | | | | | |
| White British | n (%) | 2,968 (100) | | 9,089 (97.7) | | 7,882 (93.0) | |
| Other[a] | n (%) | 0 (0) | | 205 (2.2) | | 376 (4.4) | |
| Missing | n (%) | 0 (0) | | 8 (0.1) | | 218 (2.6) | |
| **Obesity traits** | | | | | | | |
| Never obese | n (%) | 2,771 (93.4) | | 8,267 (88.9) | | 6,803 (80.3) | |
| Ever obese | n (%) | 197 (6.6) | | 1,035 (11.1) | | 1,673 (19.7) | |
| Age first onset (years) | Median (IQR) | 33.4 (27.6; 37.0) | | 31.5 (25.4; 36.1) | | 30.2 (25.2; 34.1) | |
| Total duration (years) | Median (IQR) | 6.2 (2.7; 11.8) | | 8.3 (3.9; 14.4) | | 9.7 (5.9; 14.7) | |
| Correlation (age onset × duration obese) | | −0.81 | | −0.95 | | −0.99 | |
| Number of periods | | | | | | | |
| 1 | n (%) | 192 (97.5) | | 1,023 (98.8) | | 1,671 (99.9) | |
| 2 | n (%) | 4 (2.0) | | 12 (1.2) | | 2 (0.1) | |
| 3 | n (%) | 1 (0.5) | | 0 | | 0 | |
| Obesity severity (BMI-years) | Median (IQR) | 5.6 (1.1; 22.7) | | 9.4 (1.7; 35.0) | | 17.1 (4.3; 48.0) | |
| Correlation (duration × severity) | | 0.86 | | 0.85 | | 0.80 | |
| **Biomedical outcomes** | | % missing | | % missing | | % missing | |
| Age at follow-up (years) | Mean (SD) | - | 53.5 (0.2) | - | 45.2 (0.4) | - | 47.3 (0.7) |
| BMI at follow-up (kg/m$^2$) | Median (IQR) | 1.3 | 26.6 (24.2; 29.9) | 1.3 | 26.6 (24.0; 29.9) | 13.4 | 27.6 (24.6; 31.5) |
| Obese at follow-up (BMI > 30kg/m$^2$) | n (%) | 1.3 | 707 (24.1) | 1.3 | 2,239 (24.4) | 13.4 | 2,424 (33.0) |
| Systolic blood pressure (mmHg)* | Mean (SD) | 1.9 | 136.0 (20.1) | 0.5 | 126.5 (16.5) | 11.5 | 124.6 (15.2) |
| Diastolic blood pressure (mmHg)* | Mean (SD) | 1.9 | 84.4 (12.2) | 0.5 | 78.8 (10.8) | 11.5 | 77.3 (11.0) |
| Hypertension[b] | n (%) | 1.9 | 1,993 (68.1) | 0.5 | 2,578 (27.8) | 11.3 | 1,798 (23.9) |
| HDL-C (mmol/L)* | Median (IQR) | 20.2 | 1.6 (1.3; 2.0) | 16.1 | 1.5 (1.3; 1.8) | 29.5 | 1.5 (1.2; 1.8) |
| Low-HDL-C[c] | n (%) | 19.3 | 312 (13.0) | 14.5 | 1,595 (20.1) | 28.5 | 1,385 (22.8) |
| HbA1c (%)* | Median (IQR) | 13.6 | 5.7 (5.3; 5.9) | 15.2 | 5.3 (5.0; 5.4) | 29.9 | 5.4 (5.3; 5.6) |
| Elevated HbA1c[d] | n (%) | 13.2 | 921 (35.8) | 13.8 | 1,206 (15.0) | 39.3 | 987 (16.5) |

* Original values (i.e., not adjusted for medication use).

a Other ethnicities: white other, mixed race, Indian, Pakistani, Bangladeshi, Other Asian, Caribbean, African, other black, and Chinese.

b Hypertension: SBP/DBP ≥ 140/90 mmHg and/or on BP-lowering medication.

c Low-HDL-C: according to NCEP ATPIII criteria and/or on lipid-regulating medication.

d Elevated HbA1c: according to CDC criteria and/or on diabetes medication.

BCS, 1970 British Cohort Study; BMI, body mass index; CDC, Centers for Disease Control and Prevention; DBP, diastolic blood pressure; HbA1c, glycated haemoglobin; HDL-C, high-density-lipoprotein cholesterol; NCDS, National Child Development Study; NCEP ATPIII, Adult Treatment Panel III (ATP III) report of the National Cholesterol Education Program (NCEP); NSHD, National Survey of Health and Development; SBP, systolic blood pressure.

remained ($p$(trend) = 0.007) but was attenuated, particularly for 20 to 30 years, which reduced from 19.9% to 11.6% (95% CI: 5.9, 17.2), a relative reduction of 42%.

There was also a linear trend between obesity duration and risk for elevated HbA1c, with those obese for <5 years having a 2.1 (95% CI: 1.8, 2.4) times higher risk of elevated HbA1c compared with never obese, which more than doubled in those obese for 20 to 30 years (RR 4.6; 95% CI: 3.9, 5.5, $p$(trend) < 0.001) (Fig 2, right panel). However, upon adjustment for obesity severity, this graded relationship was attenuated ($p$(trend) = 0.006).

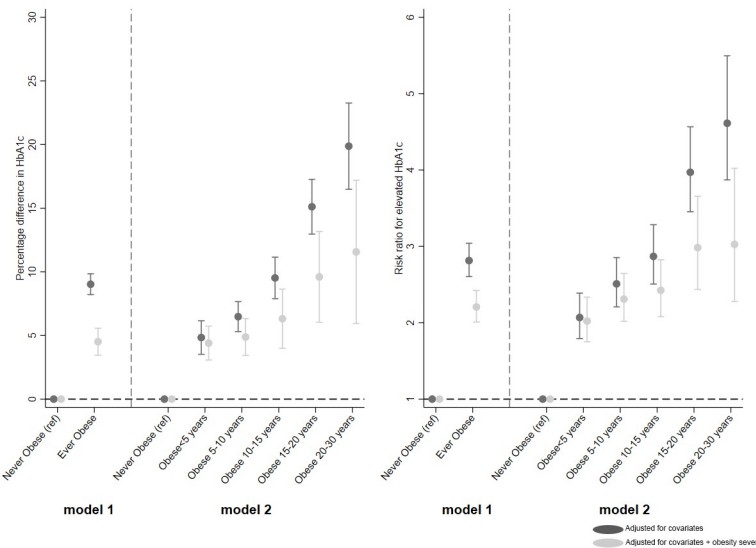

**Fig 2. Association between ever obese and categories of obesity duration (versus never obese) and HbA1c (left panel) and risk for elevated HbA1c (right panel).**

**SBP and DBP.** There was a positive relationship between ever being obese between 10 and 40 years and both SBP and DBP. For example, ever obese was associated with a 6.1% (95% CI: 5.6, 6.6) higher SBP and 7.1% (95% CI: 6.6, 7.7) higher in DBP at follow-up (versus never obese) (Fig 3A and 3B). Obesity duration was also positively associated with both SBP and DBP, such that SBP was 5.0% higher in those who were obese for <5 years compared with those never obese, increasing to 9.0% higher for 20 to 30 years ($p$(trend) < 0.001). However, upon adjustment for obesity severity, evidence for this dose-response association was greatly reduced (SBP: $p$(trend) = 0.975; DBP: $p$(trend) = 0.294).

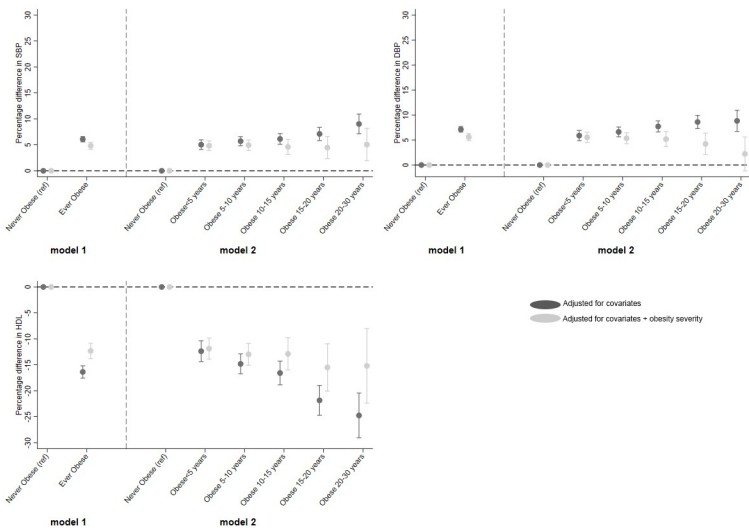

**Fig 3. Association between ever obese and categories of obesity duration (versus never obese) and SBP, DBP, and HDL-C. DBP, diastolic blood pressure; HDL-C, high-density-lipoprotein cholesterol; SBP, systolic blood pressure.**

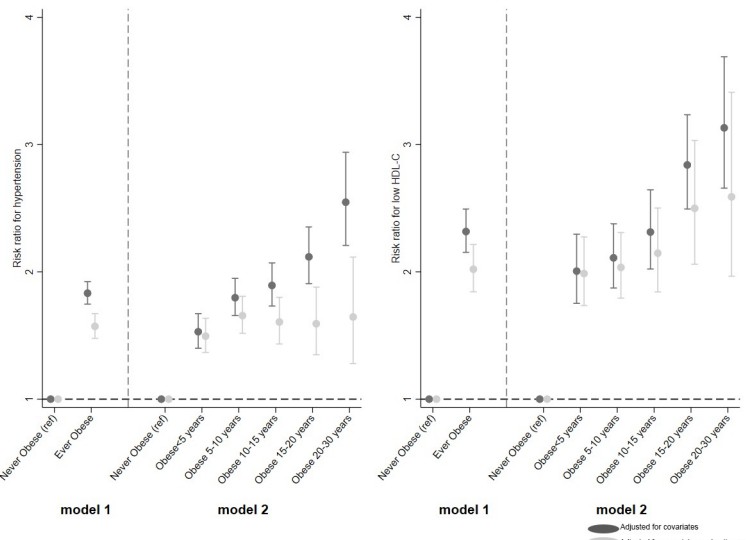

**Fig 4. Association between ever obese and categories of obesity duration (versus never obese) and risk for hypertension and low HDL-C.**

Consistent with these findings, ever being obese between 10 and 40 years (versus never) was associated with an RR for hypertension of 1.6 (95% CI: 1.5, 1.7), independent of obesity severity (Fig 4A and S6 Table). For obesity duration, a similar pattern was observed to that seen for SBP and DBP, i.e., a gradually increasing risk for hypertension with increasing time spent obese ($p$(trend) < 0.001), evidence for which weakened when adjusted for obesity severity ($p$(trend) = 0.456).

**HDL-C.** A negative relationship was observed between obesity and HDL-C, such that obesity at any point between 10 and 40 years was associated with a 16.4% (95% CI: 17.6, 15.2) lower HDL-C at follow-up (Fig 3C), attenuating to 12.3% lower when adjusted for severity. There was a linear trend in the effect of obesity duration on HDL-C, such that HDL-C levels in those with obesity for <5 years were 12.4% (95% CI: 10.4, 14.4) lower than those never obese, which increased to 24.8% (95% CI: 20.5, 29.1) lower in those who had been obese for 20 to 30 years ($p$(trend) < 0.001). Upon adjustment for obesity severity, evidence for the trend attenuated ($p$(trend) = 0.117).

This resulted in an RR for low-HDL-C of 2.0 (95% CI: 1.8, 2.2) in those who were ever obese between 10 and 40 years (versus never), independent of obesity severity (Fig 4B and S6 Table). For obesity duration, there was a linear trend of increasing risk ($p$(trend) < 0.001), which remained on adjustment for severity, although evidence for this was attenuated ($p$(trend) = 0.037).

**Sensitivity analysis.** Similar results were found when the analysis was limited to the NCDS and BCS70 cohorts (S7 and S8 Tables), thus accounting for the difference in the age at follow-up in the NSHD. Similarly, replacing the blood pressure variables in the NSHD cohort with those collected at the age 43-year sweep, in order to be more aligned with the age at follow-up in the NCDS and BCS70, did not change results (S9 and S10 Tables). When stratified by sex, associations were consistently stronger in females (S11 and S12 Tables) and especially for the dichotomous cardiometabolic disease risk factor variables. Finally, estimates were largely unchanged when the analysis was limited to those with persistent obesity (i.e., staying obese after first onset) (S13 and S14 Tables).

## Discussion

This study utilised longitudinal BMI data from 3 British birth cohort studies to model each person's obesity history and derive individual obesity parameters. Ever being obese between 10 and 40 years of age, compared with never being obese, was associated with less favourable levels of all cardiometabolic disease risk factors. More time spent obese was associated with worse profiles for all cardiometabolic disease risk factors, although greatest for HbA1c. When adjustment was made for obesity severity, the strength of the evidence in support of an association between obesity duration and SBP, DBP, and HDL-C was weak ($p > 0.1$). For HbA1c however, although the association with obesity duration also attenuated when adjusting for obesity severity, the strength of evidence remained strong. The study design, in particular the fact that most individuals who became obese remained obese, has meant that age of obesity onset and obesity duration are very highly negatively correlated. Our results also therefore mean that after accounting for obesity severity, an earlier age of obesity onset was only associated with HbA1C. These key findings were robust to a range of sensitivity analyses.

In attempting to separate the effects of obesity duration and severity on cardiometabolic health, previous studies have simply adjusted for BMI (or waist circumference) at the time of outcome assessment [14,15,19,21,22,46]. This, however, only provides an indication of obesity severity at that particular point in time. Our study represents an advance over these studies however, as we have been able to measure obesity severity accumulated over the life course, and by adjusting this for the time spent obese, we have been able to appropriately separate the effects of obesity duration and severity. As such, these findings provide novel, robust evidence regarding the independent association of obesity duration with cardiometabolic disease risk factors.

Our findings are in line with other studies, which have observed an attenuated, but persisting, effect of obesity duration on diabetes risk or impaired glucose metabolism, once obesity severity is accounted for [15,19,21,22]. In another NCDS analysis ($n = 7855$), Power and colleagues [19] observed that compared with those never obese, those with the greatest duration of obesity (i.e., onset < 16 years) had an almost 24-fold increased risk of having HbA1c of >7% (and/or a diagnosis of diabetes) at 45 years. While this risk was substantially attenuated upon adjustment for current BMI, it still remained over 4 times higher compared with those never obese. In addition we have observed, in line with Pontirolli and colleagues [15], a specific effect of obesity duration on glucose metabolism. In their study of 760 obese adults (average age 51 years), obesity duration was a risk factor for glucose intolerance and type 2 diabetes but not for hypertension or hyperlipidaemia [15]. Evidence in support of our finding of no independent association of obesity duration with HDL-C is lacking. To our knowledge only 1 other study has investigated this and observed an association in females only, although the strength of evidence was modest ($p = 0.05$) [14].

In addition to the cited empirical studies, there is also a plausible biological mechanism supporting the observed association between obesity duration and HbA1c (reflecting impaired glucose metabolism). Obesity is characterised by enlarged fat stores, which results in enhanced lipolysis and an increase in circulating free fatty acids. This state leads to peripheral and hepatic insulin resistance [47,48], resulting in a compensatory insulin hypersecretion by the pancreatic β-cells in order to preserve normoglycemia [49]. Prolonged obesity leads to β-cell exhaustion [50], culminating in a reduced insulin response and an inability to maintain normoglycemia [51]. In addition, prolonged obesity may represent a state in which subcutaneous adipose stores have been exhausted, with the consequence being a deposition of adipose tissue around the visceral organs (e.g., liver and pancreas), with fat stored in these areas (i.e., "ectopic fat") being strongly related to insulin resistance [52].

Despite the persisting independent effect of obesity duration on HbA1c levels, a substantial reduction in the effect was observed once the severity of obesity had been accounted for. This suggests that in those who have been exposed to obesity for a prolonged period, there is still opportunity to return to more favourable HbA1c levels if a degree of weight loss is achieved. For example, upon adjustment for severity, the risk of elevated HbA1c in those who had been obese for 20 to 30 years reduced from more than a 4-fold increased risk (relative to never obese), to a level similar to those obese for half as long, i.e., 10 to 15 years (RR: 3.0; 95%CI: 2.3, 4.0).

There was some evidence that the association between obesity duration and the dichotomous cardiometabolic outcomes was stronger in females than in males (S11 and S12 Tables). Sex-specific associations have been observed in other studies [14,22,53]. Janssen and colleagues [14], for example, observed an independent effect of overweight/obesity duration on risk for insulin resistance and type 2 diabetes (and also hypertension, hypertriglyceridemia, low-HDL-C, and metabolic syndrome) in females, but not in males except for hypertriglyceridemia. A sex difference was also observed in the Framingham Heart Study [53]. Tanamas and colleagues observed an association between obesity duration and risk for hypertension in females but not in males (ages 30 to 62 years) [53]. As summarised in the review by Jarvis [54], there are fundamental differences in the control of metabolic homeostasis between males and females. Females are more likely to gain fat, and although abdominal obesity more commonly affects males than females, the prevalence of abdominal obesity has increased more in females than in males [55]. Furthermore, the prevalence of visceral obesity associated with metabolic syndrome is 2 to 10 times higher in women throughout the world [56–58]. It may be therefore that compared with males, females are more exposed to this metabolically volatile adipose tissue and thus at increased risk of its deleterious outcomes.

## Strengths

The key strength of our study is the derivation, using over 130,000 serial BMI observations across the life course, of individualised obesity parameters, which enabled us to distinguish between obesity severity and duration. In addition, the pooling of data from 3 nationally representative cohorts means the observed associations are based on a far larger sample than most previous studies and are likely to be generalisable to the underlying population.

## Limitations

Our definition of obesity was based on BMI, which despite exhibiting a strong positive correlation with direct estimates of fat mass [59], is only an indicator of total body adiposity. However, it remains the most commonly used, widely accepted and practical measure of obesity in both children and adults. Our trajectories were dependent on the frequency of BMI measurements across the life course, with some intervals spanning 10 years. As such, we may not have captured instances of weight cycling between measurement occasions. Measurement protocols for weight and height were not consistent within and between studies, which may have introduced bias if self-reported measurements were systemically under or overreported. It has been shown that people with greater BMIs tend to underreport their weight [60,61], suggesting that estimates of obesity duration (and severity) may be conservative in our study. Our regression models included adjustment for only a small number of covariates, which means there is a possibility of residual confounding. As we have combined 3 cohorts, any included variable must be harmonised across each cohort so that the variable conveys the same thing in each cohort. This is only the case for a small number of variables in the cohorts we have used. As all of the included studies suffered from attrition, which is more extensive in those from lower

socioeconomic position (SEP) groups and/or with higher BMI [62,63], we may have inadvertently selected a more socioeconomically advantaged and thinner sample which, in addition to a loss of power, may have introduced bias into the observed associations. In addition, as the NSHD, NCDS, and BCS70 cohorts are either exclusively (NSHD), or predominantly white British, we are unable to generalise the results to other ethnic groups. Finally, the biomedical sweep in the NSHD cohort was conducted 9 and 7 years later than the NCDS and BCS cohorts, respectively, which may impair cross-cohort comparability (underpinning the decision to pool cohorts). However, supplementary analyses limited to the NCDS and BCS cohorts only and replacing the NSHD blood pressure variables with those collected at 43 years produced similar estimates (S7–S10 Tables).

Associations observed in this study suggest that there are benefits in delaying the onset of obesity, as risks of elevated HbA1c were positively associated with time spent with obesity, independent of the degree of severity. Interventions aiming to prevent childhood obesity therefore have the potential to reduce the long-term risk of developing diabetes. However, we also observed an amelioration of HbA1c profiles in those who had been exposed to obesity for a prolonged period, once severity of obesity is accounted for. As such, people with obesity should be encouraged to lose weight in order to return their HbA1c levels to more favourable values. Firstly, however, more research using different epidemiological approaches underpinned by different assumptions and sources of bias (e.g., mendelian randomization and negative control studies) are needed to test the robustness of these findings.

## Conclusion

We found a dose–response relationship between the duration of obesity and HbA1c, independent of obesity severity. Given that the obesity epidemic is characterised by trends towards earlier onset and consequently greater lifetime exposure, our findings are important as they suggest that health policy recommendations aimed at preventing early onset obesity, and therefore reducing lifetime obesity exposure, may help reduce the risk for diabetes. For those who are already obese, reducing obesity severity can also improve their metabolic profile. Accordingly, prevention strategies could consider both the duration and severity of obesity.

## Supporting information

**S1 Checklist. STROBE checklist.**
(DOCX)

**S1 Text. Measurement protocol for collection of cardiometabolic outcomes in adulthood.**
(DOCX)

**S2 Text. Childhood social class.**
(DOCX)

**S1 Table. Association between ever obese and categories of obesity duration (versus never obese) and cardiometabolic disease risk factors*† (imputed, unadjusted).**
(DOCX)

**S2 Table. Association between ever obese and categories of obesity duration (versus never obese) and dichotomous cardiometabolic outcomes (imputed, unadjusted).**
(DOCX)

**S3 Table. Association between ever obese and categories of obesity duration (versus never obese) and cardiometabolic disease risk factors*† (imputed, adjusted for sex, cohort, age at**

follow-up, ethnicity, birth weight, and childhood social class).
(DOCX)

**S4 Table. Association between ever obese and categories of obesity duration (versus never obese) and dichotomous cardiometabolic outcomes (imputed, adjusted for sex, cohort, age at follow-up, ethnicity, birth weight, and childhood social class).**
(DOCX)

**S5 Table. Association between ever obese and categories of obesity duration (versus never obese) and cardiometabolic disease risk factors\*† (imputed, adjusted for sex, cohort, age at follow-up, ethnicity, birth weight, childhood social class, and obesity severity).**
(DOCX)

**S6 Table. Association between ever obese and categories of obesity duration (versus never obese) and dichotomous cardiometabolic outcomes (imputed, adjusted for sex, cohort, age at follow-up, ethnicity, birth weight, childhood social class, and obesity severity).**
(DOCX)

**S7 Table. Association between ever obese and categories of obesity duration (versus never obese) and cardiometabolic disease risk factors (imputed, adjusted for sex, cohort, age at follow-up, ethnicity, birth weight, childhood social class, and obesity severity): excluding NSHD.**
(DOCX)

**S8 Table. Association between ever obese and categories of obesity duration (versus never obese) and dichotomous cardiometabolic outcomes (imputed, adjusted for sex, cohort, age at follow-up, ethnicity, birth weight, childhood social class, and obesity severity): excluding NSHD.**
(DOCX)

**S9 Table. Association between ever obese and categories of obesity duration (versus never obese) and cardiometabolic disease risk factors (imputed, adjusted for sex, cohort, age at follow-up, ethnicity, birth weight, childhood social class, and obesity severity): using blood pressure at 43 years in NSHD.**
(DOCX)

**S10 Table. Association between ever obese and categories of obesity duration (versus never obese) and categorical cardiometabolic outcomes (imputed, adjusted for sex, cohort, age at follow-up, ethnicity, birth weight, childhood social class, and obesity severity): using blood pressure at 43 years in NSHD.**
(DOCX)

**S11 Table. Association between ever obese and categories of obesity duration (versus never obese) and cardiometabolic disease risk factors (imputed, adjusted for cohort, age at follow-up, ethnicity, birth weight, childhood social class, and obesity severity): sex interaction.**
(DOCX)

**S12 Table. Association between ever obese and categories of obesity duration (versus never obese) and dichotomous cardiometabolic outcomes (imputed, adjusted for cohort, age at follow-up, ethnicity, birth weight, childhood social class, and obesity severity): sex interaction.**
(DOCX)

**S13 Table. Association between ever obese and categories of obesity duration (versus never obese) and cardiometabolic disease risk factors (imputed, adjusted for sex, cohort, age at follow-up, ethnicity, birth weight, childhood social class, and obesity severity): limited to those who once obese were always obese.**
(DOCX)

**S14 Table. Association between ever obese and categories of obesity duration (versus never obese) and dichotomous cardiometabolic outcomes (imputed, adjusted for sex, cohort, age at follow-up, ethnicity, birth weight, childhood social class, and obesity severity): limited to those who once obese were always obese.**
(DOCX)

**S1 Fig. Sample flow diagram.**
(TIFF)

## Author Contributions

**Conceptualization:** Tom Norris, Tim J. Cole, Russell Viner, William Johnson.

**Data curation:** Mark Hamer, Rebecca Hardy, George B. Ploubidis, William Johnson.

**Formal analysis:** Tom Norris, Tim J. Cole.

**Funding acquisition:** William Johnson.

**Methodology:** Tom Norris, David Bann, Mark Hamer, Rebecca Hardy, Leah Li, Ken K. Ong, George B. Ploubidis, Russell Viner, William Johnson.

**Supervision:** William Johnson.

**Writing – original draft:** Tom Norris.

**Writing – review & editing:** Tom Norris, Tim J. Cole, David Bann, Mark Hamer, Rebecca Hardy, Leah Li, Ken K. Ong, George B. Ploubidis, William Johnson.

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
