## [Editor Report · Decision Letter 0]

19 Mar 2020

Dear Dr Norris, 

Thank you for submitting your manuscript entitled "Duration of obesity exposure between ages 10-40 years and its relationship with cardiometabolic disease risk factors in 20 746 adults" for consideration by PLOS Medicine.

Your manuscript has now been evaluated by the PLOS Medicine editorial staff [as well as by an academic editor with relevant expertise] and I am writing to let you know that we would like to send your submission out for external peer review.

Kind regards,

Adya Misra, PhD,

Senior Editor

PLOS Medicine

---

## [Decision Letter · Decision Letter 1]

16 Jun 2020

Dear Dr. Norris,

Thank you very much for submitting your manuscript "Duration of obesity exposure between ages 10-40 years and its relationship with cardiometabolic disease risk factors in 20 746 adults" (PMEDICINE-D-20-00944R1) for consideration at PLOS Medicine. 

[LINK]

In light of these reviews, I am afraid that we will not be able to accept the manuscript for publication in the journal in its current form, but we would like to consider a revised version that addresses the reviewers' and editors' comments. Obviously we cannot make any decision about publication until we have seen the revised manuscript and your response, and we plan to seek re-review by one or more of the reviewers. 

We expect to receive your revised manuscript by Jul 07 2020 11:59PM. Please email us (plosmedicine@plos.org) if you have any questions or concerns.

We look forward to receiving your revised manuscript. 

Sincerely,

Emma Veitch, PhD

PLOS Medicine

On behalf of Clare Stone, PhD, Acting Chief Editor,

PLOS Medicine

plosmedicine.org

*The abstract should be modified to use the PLOS Medicine structure (Background, Methods and Findings, Conclusions) - "Methods and Findings" is a single subsection. 

*In the last sentence of the Abstract Methods and Findings section, please note any key limitation(s) of the study's methodology.

*At this stage, we ask that you include a short, non-technical Author Summary of your research to make findings accessible to a wide audience that includes both scientists and non-scientists. The Author Summary should immediately follow the Abstract in your revised manuscript. This text is subject to editorial change and should be distinct from the scientific abstract. Please see our author guidelines for more information: https://journals.plos.org/plosmedicine/s/revising-your-manuscript#loc-author-summary

*The main manuscript text should have slightly restructured headings to fit the usual PLOS Medicine style for research articles, with the main opening section headed "Introduction" and then a "Methods" (or "Methods and Materials") section.

*If possible, please reformat the in-text citation callouts to use sequential numerals in square brackets (eg, [1], [2], [3]) - if using referencing software this should be fairly straightforward and quick.

*It would be good to clarify in the paper if this analysis corresponded to an analytical plan that was prospectively planned out (eg prior to data collection/examination of data). Please state this (either way) early in the Methods section.

a) If a prospective analysis plan (from your funding proposal, IRB or other ethics committee submission, study protocol, or other planning document written before analyzing the data) was used in designing this analysis, please include the relevant prospectively written document with your revised manuscript as a Supporting Information file to be published alongside your study, and cite it in the Methods section. A legend for this file should be included at the end of your manuscript. 

Comments from the reviewers:

Reviewer #1: Comments

The adverse cardio-metabolic consequences are well known. In this paper, the authors made an effort to dissect the effects of obesity severity and obesity duration on four metabolic risk markers, namely diastolic blood pressure, systolic blood pressure, HDL-cholesterol and glycated haemoglobin (HbA1c). They utilized data collected in three British cohort studies including over 20,000 individuals. They found that a greater obesity duration was associated with worse values for all measured risk markers. The largest association with obesity duration was for HbA1c. When authors adjusted for obesity severity, the association with obesity duration was attenuated for blood pressure and HDL-cholesterol, but persisted for HbA1c. From this, they concluded that by reducing lifetime exposure to obesity may help to reduce risk of diabetes, independently of obesity severity.

Overall, I found this study well done. Below are some remarks that the authors may wish to consider:

1. Line 80. "Obesity is a global public health concern, with worldwide prevalence increasing from 100 81 million in 1975 to 671 million in 2015". 

This is could be more informative, e.g. The prevalence of obesity, defined as X, has increased from X% in 1975 to Y% in 2015 etc.

2. If venous sample was taken, it is unclear why only HbA1c and HDL-cholesterol have been analysed. This is clearly a limitation that should be explained.

3. The authors have used the International Obesity Task Force cut-points to define obesity. They may want consider performing sensitivity analyses by using more recently published cut-points from the i3C consortium (Lancet Child Adolesc Health. 2019;3:795-802).

4. The -5% constant for HDL-cholesterol seems quite high, as commonly used lipid lowering medications have a neglible effect on HDL-cholesterol.

5. The authors have not considered any covariates. Should potential confounders be taken into account, such as smoking, socioeconomic status and physical activity/sedentarity?

Reviewer #2: This is a well-analyzed and nicely written paper that will make a useful contribution to the literature. I have only a few comments:

1. Why not include triglyceride level or LDL here? Including these would strengthen the paper, if they are available.

2. Figure 3. Why not show estimates for dichotomous outcomes here as well? Dichotomizing variables is generally considered less ideal, but in this case they have the benefit of not making assumptions about the impact of medication use on continuous variables, which are certain to have individual variation. This means that both analyses will contain bias and so providing both in the main figure is useful. In addition, the dichotomized variables are tied to diagnostic criteria based on risk for cardiometabolic events and may make this paper more accessible to clinicians.

3. Did you conduct formal assessment of heterogeneity by sex?

4. Be cautious conflating a non-significant p for trend with the lack of a trend, particularly when the dichotomous outcome results differ. Especially for HDL, a marginally non-significant p for trend does not mean there is no trend, it just looks messier with more uncertainty. This is particularly questionable given the highly significant and clean trend for Low HDL. 

5. A speculation from thinking about Figure 2 and 3 that may be beyond the scope of this paper: Is duration more directly predictive of severity during the early years of obesity? This would make sense given that the development of higher levels of obesity generally takes time. Then does this mean that the heterogeneity in severity develops based on something else as duration continues? This might be an additional area of investigation, but feel free to ignore it if it makes the paper unwieldy.

Minor points: 

1. The continuous variables should not be labeled as "outcomes". These are cardiometabolic disease risk factors. The dichotomized variables (Hypertension, low HDL, high HbA1c) could more generally be referred to as "outcomes" since those cut-points are used for diagnostic purposes and are tied to cardiometabolic events like diabetes, CVD, and mortality. Although more precisely, "outcomes" is often reserved for these specific types of events. More appropriately, the continuous results would be labeled "Continuous Variables" and the dichotomized results would be labeled "Cardiometabolic Risk Factors".

2. Table 1. Providing a connection to obesity severity in a more immediately interpretable format (simple BMI) would be useful. The concept of BMI-years is not as accessible to everyone, and z-scores are notoriously hard to make intuitive and providing this in Table 1 for one of the adult assessments may help put the levels of BMI into context for readers. 

Reviewer #3: I confine my remarks to statistical aspects of this paper.

There was quite a lot to like here. The use of cubic splines to model obesity duration, for example, is excellent. But then (lines 193-196) this hard work was largely thrown away by categorizing the results. Such categorization is fine for presentation, but it weakens analysis, increasing both type I and type II error.

Instead of this, why not use obese years? Or use the results of the cubic spline analysis (perhaps clustering the trajectories - although that might be another paper). Or simply use the results of the trapezoid rule on lines 206-207?

More minor issues:

Line 66 "largest" should probably be "strongest"

Line 80-81 Please give as % of global population 

Line 84-85 I don't see how this relates to the previous sentence. Can you explain?

Line 157 What was "harmonization"? What was done and why?

Line 202-203 Don't categorize independent variables.

Peter Flom

Reviewer #4: Cole et al conducted a study attempting to evaluate the independent association of overweight duration and cardiometabolic risk factors, independently of obesity severity. This is certainly an important area of investigation as an improved understanding of the independent effect of obesity duration and severity could lead to refined public health guidance.

While the study is well-written and clearly presented, I have concerns about the methodologic approach. This relates mainly to the assessment (and modelling) of the exposure variables, the highly inter-correlated nature of the two key variables of duration and severity, and the potential for confounding by unadjusted factors. As such, I am concerned about whether the study is providing accurate estimates of the association of obesity duration and the cardiometabolic outcomes. I believe the manuscript could be improved if the authors could address the following comments:

- A major concern is residual confounding. Why were there minimal attempts to control for potential confounders - e.g. social-economic status and diet? 

- How well does the restricted cubic spline model actually capture oscillations or changes in people's BMI given the relatively long (years) interval between measurements? Could the lack of obese 'period' be a function of the way the model is fitted?

- How were the categorization cut-off values for duration of obesity chosen?

- Duration and severity very highly correlated - could the authors comment on whether multi-collinearity was an issue and if so how was it dealt with?

- How accurate are the self-reported BMI measurements? Likely to have differential misreporting by those who are overweight and obese.

Minor

- Additional text in Figure legend 1 would be helpful for readers to interpret the figure.

- Meaning of line 223-224 unclear: 'data were pooled across cohorts and sexes, enabling adjustment of the association between obesity duration and outcomes for these variables'.

- Line 265 - remove the word 'people'.

- Line 271 - should provide correlation for NCDS too.

[LINK]

---

## [Decision Letter · Decision Letter 2]

21 Aug 2020

Dear Dr. Norris,

Thank you very much for re-submitting your manuscript "Duration of obesity exposure between ages 10-40 years and its relationship with cardiometabolic disease risk factors in 20 746 adults" (PMEDICINE-D-20-00944R2) for review by PLOS Medicine.

I have discussed the paper with my colleagues and the academic editor and it was also seen again by two of the original reviewers. I am pleased to say that provided the remaining editorial and production issues are dealt with we are planning to accept the paper for publication in the journal.

[LINK]

We look forward to receiving the revised manuscript by Aug 28 2020 11:59PM. 

Sincerely,

Thomas McBride, PhD

Senior Editor 

PLOS Medicine

plosmedicine.org

Requests from Editors:

1-Please ensure that the study is reported according to the STROBE guideline, and include the completed STROBE checklist as Supporting Information. 1 Please add the following statement, or similar, to the Methods: "This study is reported as per the Strengthening the Reporting of Observational Studies in Epidemiology (STROBE) guideline (SChecklist)."

2- Thank you for noting when the analysis plan was determined. Please also note when in relation to seeing the data analyses were planned, as well as any changes that were made to the analysis plan after seeing the data (e.g. your response to Reviewer 2, point 1 regarding removing triglycerides from the risk factors; and your response to Reviewer 3, point 1 regarding obese years), and any new analyses conducted in response to reviewer comments.

3- Please revise your title according to PLOS Medicine's style. Your title must be nondeclarative and not a question. It should begin with main concept if possible. Please place the study design ("A randomized controlled trial," "A retrospective study," "A modelling study," etc.) in the subtitle (ie, after a colon). 

Suggestion: “Duration of obesity exposure between ages 10-40 years and its relationship with

cardiometabolic disease risk factors: a cohort study”

4- Abstract Conclusions: Please address the study implications without overreaching what can be concluded from the data. Perhaps better to suggest it is worth exploring the potential link between obesity duration and biomarkers of diabetes. It is also worth mentioning the negative findings here.

5- As with the Abstract conclusions, please edit the “What Do These Findings Mean?” section of the Author Summary.

6- Thank you for noting that the initial studies received ethical approval and participant informed consent, please also note if such approval was sought for the current study or if these analyses were covered by the initial approval and consent.

7- Line 306: Thank you for agreeing to provide the statistical code. Instead of requiring readers to contact the corresponding author, please place the code in a repository (such as GitHub, SourceForge or Bitbucket) or a cloud computing service (such as Code Ocean). Protection of authors’ intellectual property will not be cause for exception. Please explain in the manuscript’s Data Availability Statement how readers can access the shared code. 

8- Please include ethnicity when describing the participant demographics in the Abstract, Table 1, and at thes start of the Results section. If the cohorts are close to 100% white, please note this as a limitation.

9- At line 392, rather than “disappeared”, perhaps say that “... the association with obesity duration was attenuated and no longer significant…”

10- Similarly, at line 393, “For HbA1c however, though the association with obesity duration also was attenuated when adjusting for obesity severity, it remained significant.”

11- Please rename the “Conclusions” Section as “Discussion”.

12- In the Discussion, between the Limitations and the Conclusion sections, please include a paragraph on the implications and next steps for research, clinical practice, and/or public policy.

13- Please include a list of all supplemental files, with titles and legends, at the end of the main text.

14- Please place a space before the brackets for the reference call-outs.

15- Reference 31: you can remove “Lehman R, editor. ” from the citation. PLOS Medicine reference guidelines are here: https://journals.plos.org/plosmedicine/s/submission-guidelines#loc-references

Comments from Reviewers:

Reviewer #1: No further comments.

[LINK]

---

## [Editor Report · Decision Letter 3]

1 Oct 2020

Dear Dr Norris, 

On behalf of my colleagues and the academic editor, Dr. Jason Wu, I am delighted to inform you that your manuscript entitled "Duration of obesity exposure between ages 10-40 years and its relationship with cardiometabolic disease risk factors: a cohort study" (PMEDICINE-D-20-00944R3) has been accepted for publication in PLOS Medicine. 

PRODUCTION PROCESS

Before publication you will see the copyedited word document (within 5 busines days) and a PDF proof shortly after that. The copyeditor will be in touch shortly before sending you the copyedited Word document. We will make some revisions at copyediting stage to conform to our general style, and for clarification. When you receive this version you should check and revise it very carefully, including figures, tables, references, and supporting information, because corrections at the next stage (proofs) will be strictly limited to (1) errors in author names or affiliations, (2) errors of scientific fact that would cause misunderstandings to readers, and (3) printer's (introduced) errors. Please return the copyedited file within 2 business days in order to ensure timely delivery of the PDF proof. 

If you are likely to be away when either this document or the proof is sent, please ensure we have contact information of a second person, as we will need you to respond quickly at each point. Given the disruptions resulting from the ongoing COVID-19 pandemic, there may be delays in the production process. We apologise in advance for any inconvenience caused and will do our best to minimize impact as far as possible.

PRESS

PROFILE INFORMATION

Thank you again for submitting the manuscript to PLOS Medicine. We look forward to publishing it. 

Best wishes, 

Thomas McBride, PhD

Senior Editor 

PLOS Medicine

plosmedicine.org